# Media framing of childhood obesity: a content analysis of UK newspapers from 1996 to 2014

Amy Nimegeer, Chris Patterson, Shona Hilton

MRC/CSO Social & Public Health Sciences Unit, University of Glasgow, Glasgow, UK

**Correspondence to**
Dr Amy Nimegeer;
Amy.Nimegeer@glasgow.ac.uk

## ABSTRACT

**Background**  Media can influence public and policy-makers' perceptions of causes of, and solutions to, public health issues through selective presentation and framing. Childhood obesity is a health issue with both individual-level and societal-level drivers and solutions, but public opinion and mass media representations of obesity have typically focused on individual-level framings, at the cost of acknowledgement of a need for regulatory action.

**Objective and setting**  To understand the salience and framing of childhood obesity across 19 years of UK national newspaper content.

**Design and outcome measures**  Quantitative content analysis of 757 articles about childhood obesity obtained from six daily and five Sunday newspapers. Articles were coded manually for definitions, drivers and potential solutions. Data were analysed statistically, including analysis of time trends and variations by political alignment of source.

**Results**  The frequency of articles grew from a low of two in 1996 to a peak of 82 in 2008, before declining to 40 in 2010. Individual-level drivers (59.8%) and solutions (36.5%) were mentioned more frequently than societal-level drivers (28.3%) and solutions (28.3%) across the sample, but societal solutions were mentioned more frequently during the final 8 years, coinciding with a marked decline in yearly frequency of articles.

**Conclusions**  Increased focus on societal solutions aligns with public health goals, but coincided with a reduction in the issue's salience in the media. Those advocating public policy solutions to childhood obesity may benefit from seeking to raise the issue's media profile while continuing to promote structural conceptualisations of childhood obesity.

## Strengths and limitations of this study

► Methodology includes systematic analysis of a large sample of 19 years of UK national newspaper coverage, facilitating statistical understandings of media frames of childhood obesity, including definitions, drivers and solutions.

► Features robust manual coding and links to pre-existing dataset to strengthen analysis.

► Quantitative media content analysis is inherently less sensitive to nuance than qualitative analysis, and our analysis excluded some aspects of media content, such as images, which may influence readers' interpretations of the text they accompany.

► Content analysis is a means of documenting what messages are presented by media, but cannot tell us how these messages are received by audiences.

## INTRODUCTION

Childhood obesity has been described as an international epidemic due to its high prevalence and rapid growth in numerous countries.[1] Globally, 13.4% of girls and 12.9% of boys in low-income countries, and 22.6% of girls and 23.8% of boys in high-income countries (classified by the World Bank) were classified as overweight or obese in 2013.[2] In England, one-fifth of children in reception year (age 4–5), and one-third in year 6 (age 10–11), were classified as overweight or obese

in 2015/2016.[3] In Scotland, 28% of children aged 2–15 were classified as 'at risk of' overweight or obesity in 2015.[4] Childhood obesity has a broad range of short-term and long-term health consequences,[1] tends to predict adolescent and adult obesity,[5] and is socio-economically patterned.[3] For these reasons, childhood obesity has been identified as a health priority for the UK and its devolved governments.[6 7]

Childhood obesity is a complex problem, with a complex set of drivers and potential solutions ranging from the individual to the environmental.[8] Ebbeling *et al*[1] identify a wide range of causes, but argue that the problem 'can be primarily attributed to adverse environmental factors', and identify a need for 'straightforward, if politically difficult' solutions spanning homes, schools, the built environment, healthcare, marketing, media and politics. This multilevel package of solutions echoes Friedman's assertion that a 'full-court press' targeting 'every dimension of the problem' is necessary.[9] However, while academia and public health are united on the need to target the obesogenic environment, Swinburn *et al*[10] state that 'governments have largely abdicated the responsibility for

addressing obesity to individuals, the private sector and non-governmental organisations',[10] potentially due to anticipated or actual resistance, not just from corporations, but also electorates.[9 11] Indeed, public opinion research conducted in the USA[12] and Germany[13] suggests that, while public are in favour of tackling childhood obesity, they demonstrate less enthusiasm for regulative environmental interventions such as taxation. Hilbert *et al* characterise the German population as 'ready for obesity prevention', but in need of education about the definition, prevalence and causes of obesity.[12]

The media represent a key influence on public perceptions of health issues and policies, setting the public agenda by granting differing levels of prominence to different topics[14] and influencing how those issues are understood by building frames (focuses of attention) that include constructions of problems, affected groups, drivers and solutions.[15 16] The influence of framing is well established in relation to obesity. Researchers have used experimental designs to demonstrate that 'individualised' representations of childhood obesity tend to lead participants to assign greater blame to individuals and exhibit less support for environmental regulation,[17] and that different representations of the consequences of childhood obesity can influence participants' attitudes towards policies.[18] Similarly, Barry *et al* demonstrated that people's perceptions of obesity (as communicated through agreement with metaphor-based descriptions of obesity) predict their support for public policy interventions, illustrating how, for example, framing obesity as being driven by industry manipulation may lead to increased support for a 'junk-food tax'.[19]

The media are frequently accused of contributing to obesity, particularly childhood obesity, through its associations with sedentary behaviour, advertising of unhealthy commodities, promotion of unrealistic body image and other mechanisms.[20] Many researchers have studied media representations of obesity in general,[21–33] but relatively few have focused specifically on representations of childhood obesity, and these have been primarily in the USA and Australia. Barry *et al*[34] studied US print and television news framing of childhood obesity, observing that coverage of the issue grew between 2000 and 2009, and that individual-level behavioural solutions to obesity were dominant, particularly on television. Similarly, Hawkins and Linvill[35] studied US newspaper framing of childhood obesity over three discrete time periods in 1991, 2001 and 2006, and identified a predominant focus on individual-level factors (both individual children and their parents) in representations of both causes and solutions. Bastian[36] analysed representations of childhood obesity in both Australian newspapers and academic literature in 2009, identifying predominantly individual framing within the media, compared with a social structural framing in academic literature. Bastian[36] recommends that public health professionals work to redirect media attention towards structural drivers of childhood obesity. Maher *et al*[37] analysed constructions of maternal

responsibilities within Australian media coverage of childhood obesity, concluding that the dominant framing 'individualises maternal and child relationships rather than seeing mothering as embedded in broader social and economic structures', serving a neoliberal agenda by diminishing the responsibility of wider society. This is consistent with the disproportionate focus on individual-level solutions identified by others.[34–36 38] While coverage of obesity in both adults and children appears to be characterised by individual-framing, it is notable that with adult obesity that individual responsibility is assigned to the person with obesity, while in childhood obesity that responsibility is predominantly assigned to parents, particularly mothers.[39 40] This distinction may complicate direct comparison between adult and child obesity, and the culturally ingrained nature of the concept that parents (or mothers) are solely responsible for their children's healthcare may represent a discursive obstacle to attempts to assign environmental solutions to childhood obesity.

In addition to traditional news media, researchers have analysed representations of childhood obesity in non-news media and new media. For example, Kalin and Fung's[38] analysis of Spanish-language US parenting magazines' representations of childhood obesity prevention and control echoes studies of news media representations of obesity, identifying greater focus on parental behavioural change than system-level solutions, and limited recognition of social contextual factors. In recognition of the growing importance of user-generated social content and discussion, researchers have increasingly analysed content about childhood obesity on social media platforms.[41 42] While these new forms of media content represent an important aspect of the changing media landscape, traditional media outlets remain influential; despite the precipitous decline of UK print newspaper circulation,[43] the online presences of these hegemonic print news brands largely dominate online news readership,[44] and typically define or legitimise news agendas for social media discussion.[45 46] However, it is also true that the relationship between news media and social media is interconnected and complex: social media trends are likely to influence the salience granted to issues by mainstream media outlets; social media posts frequently find themselves the object of news media reporting; and readers' comments on online traditional news articles can form part of the 'text' for subsequent readers. As an integral part of this complex new landscape, traditional media remain a relevant subject for media analysis, particularly when studying how representations evolve over time frames predating the ascendancy of new media.

The aim of this study is to further understandings of media representations of childhood obesity in the UK context, using an approach informed by media framing theory,[15 16] analysing definitions of the problem and constructions of drivers and solutions. This is

important because, while childhood obesity in the UK shares many similarities with that of other countries, the UK context differs in terms of several elements including health service structure and media environment. The analysis will have dual foci: the evolution of coverage between 1996 and 2014, and the relative salience of individual and societal constructions of the drivers of, and potential solutions to, childhood obesity. To our knowledge, this research will be the first empirical analysis of UK media framing of childhood obesity. This paper comprises the UK portion of a multicountry research project, the other parts of which will be reported in separate papers.

## METHODS

The media content analysis methods used were predominantly based on Hilton *et al* prior study[21] of UK newspaper framing of obesity in the general population, adapted for this study's focus on childhood obesity. This paper reports UK data that was part of a wider study that examined childhood obesity media coverage in two other international contexts; Sweden and the USA. Although content analysis is often viewed as an objective, descriptive approach, we subscribe to Krippendorff's position that even the quantitative analysis of text is inherently an interpretive act, and researchers should, therefore, acknowledge the individual bias that can arise from that process, seeking to minimise that bias through research design, while also embracing how researchers' contextual understandings can enrich coding and analysis beyond the crude 'objective' counting of content.

### Patient and public involvement

Due to the nature of this study, patients/public were not involved.

### Sampling

A set of six daily newspapers and five Sunday newspapers with high circulation figures[47] and representing a variety of political alignments and markets (or 'genres') were chosen. Table 1 lists these publications and indicates their political alignments and the markets that they occupy. Markets were defined as tabloid (typically sensationalist and politically diverse, with predominantly working-class readerships), middle-market tabloid (centre-right content with predominantly older, middle-class readerships) and quality (serious tone with predominantly middle-class readerships), using a typology used in prior studies of UK newspaper content.[21] Political alignment was determined by cross-referencing data on: the political party endorsed by each publication at the 2017 UK general election[48]; readers' perceptions of newspapers' political alignment[49] and the voting behaviours of each publications' readers in the 2015 UK general election.[50] A sample period of 1996–2014 was chosen to encompass the time period covered in prior research,[21] in addition to a further 4 years of coverage that was extended to align with the time period covered by the other countries in our wider study (which will be described fully in a separate publication).

Identifying relevant articles from the chosen publications involved an initial database search, followed by manual filtering of search results. The *Nexis* database was searched for the presence of both the term 'obesity' OR 'obese' OR 'fat' and the term 'child' OR 'children' OR 'kid' OR kids' within the headlines of articles published within the selected newspapers. Each chosen publication was archived comprehensively within the *Nexis* database, with the exceptions of the *Daily Telegraph and Sunday Telegraph* prior to October 2000 and November 2000, respectively. As such, reporting from those publications during the first 5 years of the sample period was not represented. The initial search returned 1199 articles, which were subsequently subjected to manual application of exclusion criteria, including: less than 50% of article content focussing on childhood obesity (ie, where more than half the article discussed another topic with only brief mention of childhood obesity);

**Table 1** Summary of article characteristics

| Publication | Political alignment | Market | All articles n (%)* | Front-page articles n (%)† | Word count First quartile | Median | Third quartile |
|---|---|---|---|---|---|---|---|
| *Guardian* and *Observer* | Centre left | Quality | 109 (14.4) | 5 (4.6) | 457 | 680 | 907 |
| *Independent* and *Independent on Sunday* | Centre left | Quality | 61 (8.1) | 0 | 247 | 474 | 690 |
| *Mirror* and *Sunday Mirror* | Centre left | Tabloid | 198 (26.2) | 2 (1.0) | 121 | 219 | 459 |
| *Daily Telegraph* and *Sunday Telegraph* | Centre right | Quality | 107 (14.1) | 9 (8.4) | 182 | 346 | 502 |
| *Daily Mail* and *Mail on Sunday* | Centre right | Middle market | 134 (17.7) | 6 (4.5) | 263 | 438 | 672 |
| *Sun* | Centre right | Tabloid | 148 (19.6) | 0 | 98 | 195 | 337 |
| Total | | | 757 (100.0) | 22 (2.9) | 151 | 325 | 595 |

*Percentage within whole sample.
†Percentage of front-page articles within publication.

being a reader's letter; or being part of television guide section. Following exclusion, the final sample comprised 757 relevant articles.

## Coding

Article content was coded quantitatively using a coding frame adapted from one initially developed by Hilton *et al*.[21] The adapted coding frame was developed to record media frames of childhood obesity in terms of definitions of the problem, mentions of specific biological, individual and societal drivers, and biological, individual and societal solutions (itemised in table 2). In addition, the coding frame recorded whether the article was published on the front page of the publication and the length of the article in number of words. Articles were coded as relating to women/girls or men/boys if members of that gender were described as being specifically problematic in relation to childhood obesity (but not if rates for both genders were cited), or if the article profiled an individual of a specific gender. Coding was performed by AN and CP, and 10% of articles were double-coded blind to allow inter-rater agreement to be calculated. Cohen's kappa values for agreement on individual codes are listed in table 2. The threshold for acceptable agreement was set at 0.61 (defined by Landis and Koch as 'substantial' or better agreement),[51] and three codes were removed due to insufficient agreement: dieting (such as fad diets) as a driver of childhood obesity; normalisation of obesity as a driver of childhood obesity; and technological developments as a driver of childhood obesity.

## Analysis

Statistical analysis was performed in STATA V.14.2. Statistical procedures included: basic descriptive statistics; Cohen's kappa test of inter-rater agreement; linear regression of relationships between publication year and mentions of different categories of drivers and solutions; and multiple logistic regression of relationships between political alignment and individual aspects of framing. The multiple logistic regressions were adjusted by publication market because the markets represented were not distributed evenly by political alignment (as is the case in the UK newspaper industry), and previous research has identified significant variation in health news coverage by publication market.[21 52 53]

## Comparative analysis

Data from Hilton *et al*'s previous study on representations of general (not childhood-specific) obesity in the UK media were also analysed which had been collected and described fully elsewhere[21] to enable comparison of newspaper representations of obesity in children with obesity in adults and obesity coverage more generally. This direct comparison was enabled by the intentional similarity of the methods of data collection, coding and analysis in the two studies.

**Table 2** Frequency of mentions of problem definitions, drivers, and categories of solutions

| Theme | Total (n=757) n (%) | Inter-rater agreement* |
|---|---|---|
| **Problem definitions** | | |
| Quantifies obesity prevalence within the UK | 413 (54.6) | 0.834 |
| Quantifies obesity prevalence elsewhere | 80 (10.6) | 0.814 |
| Mentions increase in obesity rates | 389 (51.4) | 0.940 |
| Mentions obesity as a risk to health | 397 (52.4) | 0.893 |
| Mentions obesity as a cosmetic problem | 23 (3.0) | 0.850 |
| Mentions obesity as a burden to NHS | 102 (13.5) | 0.814 |
| Mentions obesity as an economic burden to society | 32 (4.2) | 0.630 |
| Mentions socioeconomic and geographical differences | 74 (9.8) | 0.706 |
| Mentions women and/or girls | 112 (14.8) | 0.706 |
| Mentions men and/or boys | 56 (7.4) | 0.706 |
| Obesity is not a problem, overhyped | 93 (12.3) | 0.850 |
| Mentions discrimination, bullying or stigmatisation | 70 (9.2) | 1.000 |
| **Drivers of obesity** | | |
| **Overall drivers** | | |
| Any drivers mentioned | 522 (69.0) | N/A† |
| Any biological/genetic driver mentioned | 70 (9.2) | N/A† |
| Any individual driver mentioned | 453 (59.8) | N/A† |
| Any societal driver mentioned | 214 (28.3) | N/A† |
| **Individual drivers** | | |
| Mentions poor diet, overeating | 235 (31.0) | 0.857 |
| Mentions poor self-control, willpower or choices | 60 (7.9) | 0.680 |
| Mentions insufficient exercise, sedentary lifestyle | 224 (29.6) | 0.919 |
| Mentions parenting shortcomings | 246 (32.5) | 0.939 |
| **Societal drivers** | | |
| Mentions an abundance of processed/fast food | 129 (17.0) | 0.752 |
| Mentions a lack of health services or facilities | 53 (7.0) | 0.945 |
| Mentions food/drink advertising and promotions | 90 (11.9) | 1.000 |
| **Solutions to obesity** | | |
| Any solution mentioned | 538 (71.1) | N/A† |
| Individual solution mentioned | 276 (36.5) | 0.920 |
| Societal solution mentioned | 214 (28.3) | 0.839 |
| Biological solution mentioned | 52 (6.9) | 1.000 |

*Cohen's kappa test of inter-rater agreement.
†Agreement was not calculated for these variables as they were computed from other, manually coded variables.
N/A, not applicable; NHS, National Health Service.

## RESULTS

## Sample characteristics

Table 1 summarises the political alignment and market of each publication in the sample, in addition to the frequency of articles and front-page articles within those publications, and the variation in word count within those articles. A total of 757 articles relevant to childhood obesity

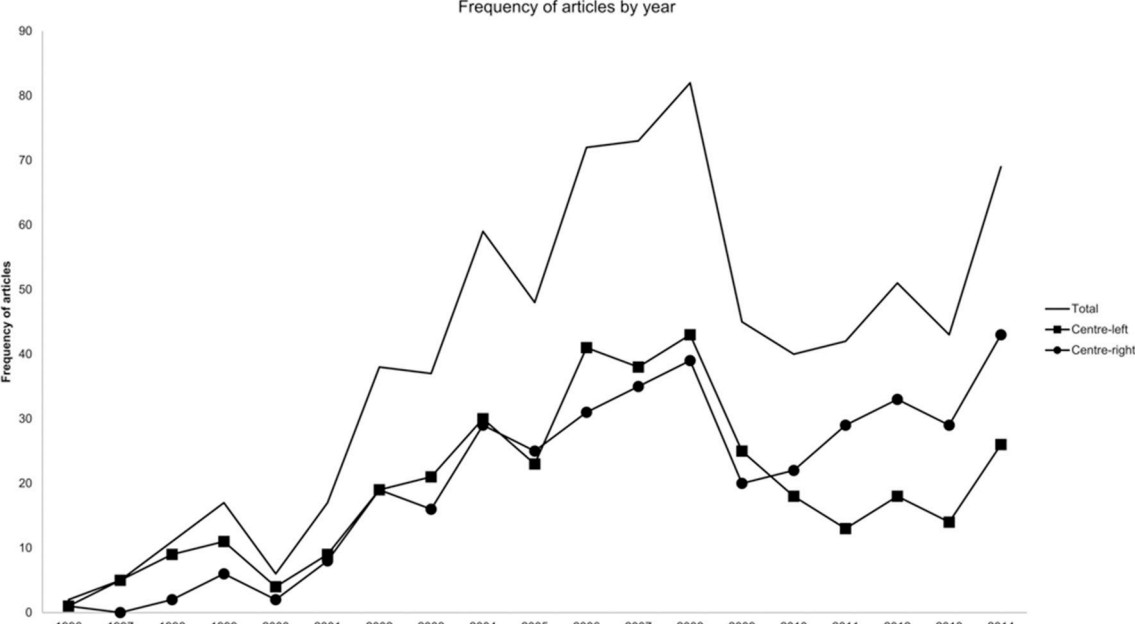

**Figure 1** Frequency of articles by year.

were identified within the selected six publications (five of which were combined with their corresponding Sunday counterparts). The frequency of coverage of childhood obesity varied between publications, ranging from the *Independent* and *Independent on Sunday* publishing 61 relevant articles, none of which were on front pages, to the *Mirror* and *Sunday Mirror*, which published 198 relevant articles, including two front-page articles. The *Daily Telegraph* and *Sunday Telegraph* afforded the issue the greatest prominence, featuring it on their front pages nine times.

The changing frequency of relevant articles within the sample between 1996 and 2014 is illustrated in figure 1, both overall and within each political alignment. The total number of relevant articles per year rose steadily from 2 in 1996 to a high of 82 in 2008, before declining to 40 in 2010, and finally rising again to 69 articles in 2014. The peak from 2006 to 2008 was contemporaneous with the publication of the UK Government's Foresight project report on reducing obesity[8] and its corresponding mid-term and 1-year reviews.

### Definitions of the problem of childhood obesity
Table 2 illustrates the frequencies of articles mentioning specific problem definitions, drivers and solutions related to childhood obesity, and table 3 illustrates the extent to which publications' political alignment predicted mentions of specific definitions. More than half of articles quantified childhood obesity prevalence within the UK (n=413, 540.6%), and a similar proportion described obesity prevalence as rising, or having risen (n=389, 51.4%). Centre-right-aligned publications mentioned increasing prevalence significantly less frequently than centre-left publications (OR 0.59; p=0.001). Eighty (10.6%) articles quantified the prevalence of obesity outside of the UK. Approximately half of articles specifically described obesity as a health risk (n=397, 52.4%)

and 102 (13.5%) described it as a burden to the National Health Service, and each of these themes was more frequent in centre-left publications (OR 0.35, p=0.010; OR 0.50, p=0.008).Childhood obesity was characterised as an economic burden to society in 74 (9.8%) articles, and significantly more so in centre-left publications (OR 0.35, p=0.010).

Few articles (n=23, 3.0%) characterised obesity as a cosmetic problem. Twice as many articles mentioned childhood obesity in relation to women and/or girls (n=112, 14.8%) as men and/or boys (n=56, 7.4%), and men and/or boys were more likely to be mentioned in centre-left publications than centre-right publications, after adjusting for market (OR 0.43, p=0.020).

### Presentations of potential drivers of, and solutions to, childhood obesity
Mentions of specific drivers of childhood obesity were coded and categorised as either individual (n=453, 59.8%), societal (n=214, 28.3%) or biological/genetic (n=70, 9.2%) drivers (table 2). Societal drivers were mentioned more frequently in centre-left publications (OR 0.69, p=0.046). Frequently mentioned individual drivers included parenting (n=246, 32.5%), diet (n=235, 31.0%) and insufficient exercise (n=224, 29.6%), while societal drivers included an abundance of unhealthy food (n=129, 17.0%), marketing (n=90, 11.9%) and insufficient health services or facilities (n=53, 7.0%).

In addition to drivers, mentions of potential solutions to childhood obesity were coded into three corresponding categories: individual (n=276, 36.5%), societal (n=214, 28.3%) and biological (n=52, 6.9%) (table 2). Table 4 illustrates the extent to which publications' political alignment predicted mentions of specific drivers and solutions. After adjusting for publication market, centre-left publications were more likely to mention societal

**Table 3** Likelihood of centre-right-aligned publications mentioning definitions of obesity

| | Unadjusted | | | Adjusted* | | |
|---|---|---|---|---|---|---|
| | OR | 95% CI | P value | OR | 95% CI | P value |
| **Problem definitions** | | | | | | |
| Quantifies obesity prevalence within the UK | 0.97 | 0.73 to 1.30 | 0.858 | 0.92 | 0.67 to 1.27 | 0.608 |
| Quantifies obesity prevalence elsewhere | 0.63 | 0.40 to 1.01 | 0.057 | 0.59 | 0.34 to 1.03 | 0.065 |
| Mentions increase in obesity rates | 0.70 | 0.52 to 0.93 | **0.014** | 0.59 | 0.42 to 0.81 | **0.001** |
| Mentions obesity as a risk to health | 1.02 | 0.77 to 1.36 | 0.885 | 0.88 | 0.64 to 1.22 | 0.456 |
| Mentions obesity as a cosmetic problem | 0.40 | 0.16 to 0.99 | **0.048** | 0.35 | 0.11 to 1.05 | 0.061 |
| Mentions obesity as a burden to NHS | 0.57 | 0.37 to 0.87 | **0.009** | 0.50 | 0.30 to 0.83 | **0.008** |
| Mentions obesity as an economic burden to society | 0.36 | 0.19 to 0.70 | **0.003** | 0.35 | 0.16 to 0.78 | **0.010** |
| Mentions socioeconomic and geographical differences | 0.62 | 0.38 to 1.00 | 0.051 | 0.85 | 0.51 to 1.43 | 0.547 |
| Mentions women and/or girls | 0.86 | 0.58 to 1.29 | 0.467 | 0.77 | 0.48 to 1.23 | 0.271 |
| Mentions men and/or boys | 0.59 | 0.34 to 1.03 | 0.062 | 0.43 | 0.22 to 0.88 | **0.020** |
| Obesity is not a problem, overhyped | 0.75 | 0.29 to 1.93 | 0.552 | 0.73 | 0.25 to 2.15 | 0.565 |
| Mentions discrimination, bullying or stigmatisation | 0.56 | 0.36 to 0.87 | **0.010** | 0.44 | 0.25 to 0.76 | **0.003** |

Bold type denotes statistical significance defined as p<0.05.
*Adjusted for publication market.
NHS, National Health Service.

drivers (OR 0.69, p=0.046) and societal solutions (OR 0.54, p=0.000). Regarding specific societal drivers, centre-left publications were more likely to mention marketing (OR 0.55, p=0.030) the an abundance of fast food (OR 0.61, p=0.011), but the latter was only significant before adjusting for publication market.

**Table 4** Likelihood of centre-right-aligned publications mentioning categories of driver and solution

| | Unadjusted | | | Adjusted* | | |
|---|---|---|---|---|---|---|
| | OR | 95% CI | P value | OR | 95% CI | P value |
| **Drivers of obesity** | | | | | | |
| Overall drivers | | | | | | |
| Any drivers mentioned | 0.90 | 0.66 to 1.23 | 0.505 | 0.78 | 0.56 to 1.10 | 0.162 |
| Any biological/genetic driver mentioned | 0.73 | 0.45 to 1.20 | 0.214 | 0.85 | 0.49 to 1.46 | 0.557 |
| Any individual driver mentioned | 1.00 | 0.75 to 1.34 | 0.974 | 0.84 | 0.61 to 1.16 | 0.292 |
| Any societal driver mentioned | 0.62 | 0.45 to 0.86 | **0.004** | 0.69 | 0.48 to 0.99 | **0.046** |
| Individual drivers | | | | | | |
| Mentions poor diet, overeating | 0.73 | 0.54 to 0.99 | **0.045** | 0.65 | 0.46 to 0.93 | **0.018** |
| Mentions poor self-control, willpower or choices | 0.61 | 0.35 to 1.04 | 0.068 | 0.71 | 0.39 to 1.28 | 0.255 |
| Mentions insufficient exercise, sedentary lifestyle | 0.75 | 0.55 to 1.03 | 0.077 | 0.67 | 0.47 to 0.97 | **0.032** |
| Mentions parenting shortcomings | 1.14 | 0.84 to 1.55 | 0.386 | 1.08 | 0.77 to 1.52 | 0.660 |
| Societal drivers | | | | | | |
| Mentions an abundance of processed/fast food | 0.61 | 0.41 to 0.89 | **0.011** | 0.73 | 0.48 to 1.12 | 0.153 |
| Mentions a lack of health services or facilities | 0.90 | 0.52 to 1.58 | 0.725 | 0.87 | 0.46 to 1.65 | 0.671 |
| Mentions food/drink advertising and promotions | 0.56 | 0.36 to 0.88 | **0.012** | 0.55 | 0.32 to 0.94 | **0.030** |
| **Solutions to obesity** | | | | | | |
| Biological | 0.73 | 0.42 to 1.29 | 0.286 | 0.54 | 0.26 to 1.09 | 0.087 |
| Individual | 0.90 | 0.67 to 1.20 | 0.464 | 0.90 | 0.64 to 1.25 | 0.527 |
| Societal | 0.62 | 0.46 to 0.83 | **0.001** | 0.54 | 0.39 to 0.75 | **0.000** |

Bold type denotes statistical significance defined as p<0.05.
*Adjusted for publication market.

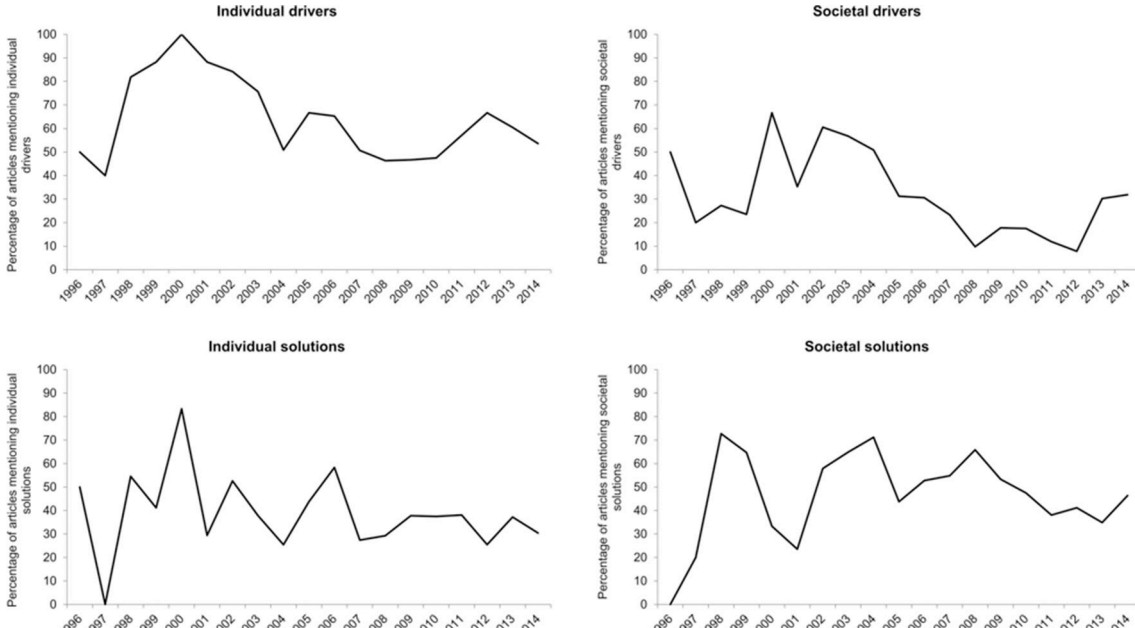

**Figure 2** Trends in individual and societal drivers and solutions.

## Time trends in presentation of drivers and solutions

Time trends in mentioning each category were analysed. Mentions of individual drivers (coefficient −0.068, p<0.001), individual solutions (coefficient −0.037, p=0.041), societal drivers (coefficient −0.097, p<0.001) and societal solutions (coefficient −0.044, p=0.012) each decreased significantly between 1996 and 2014. Neither biological/genetic drivers (coefficient −0.014, p=0.637) nor biological solutions (coefficient −0.020, p=0.558) varied significantly across the sample period.

Figure 2 illustrates the trends in individual and societal drivers and solutions. Individual drivers were mentioned particularly frequently (82%–100%) between 1998 and 2000, before declining to between 46% and 67% of articles between 2004 and 2014. Mentions of individual solutions peaked at 83% in 2000, and subsequently declined, comprising 25%–38% articles between 2007 and 2014. Mentions of societal drivers peaked at 67% in 2000, followed by a lower peak of 61% in 2002 and a subsequent lengthy decline to a low of 8% in 2012. Mentions of societal solutions exhibited a less linear decline than other categories, with peaks in 1998 (73%), 2004 (71%) and 2008 (66%), interspersed with declines. Notably, societal solutions were more commonly mentioned than individual solutions from 2007 to 2014.

## Representations of childhood obesity in comparison to adult and general population obesity

The data collected for this study were compared with data collected in Hilton *et al* 2012 study of newspaper representations of obesity in the general population.[21] Supporting information online supplementary figure S1 illustrates the yearly frequency of articles in each study's main sample, as well as a subsample of the previous study's data that excludes all articles that mentioned children. This represents a means of comparing representations of childhood obesity with representations of adult and non-age-specific obesity. Online supplementary figure S1 suggests that childhood obesity received less newspaper coverage than adult obesity in every year covered by the two datasets, with the exception of 1999. The longer time period represented in the current study suggests that the decrease in publication frequency in 2008–2010 observed in the previous study[21] did not continue in subsequent years, at least on the topic childhood obesity.

Supporting information online supplementary figure S2 illustrates the frequency of coverage of the two categories of driver and solution, individual and societal, within the present and prior sample.[21] Comparison of the data indicates that coverage of childhood obesity was characterised by greater focus on individual drivers and societal solutions than coverage of adult obesity, while coverage of societal drivers and individual solutions was relatively similar.

## DISCUSSION

By systematically analysing the content of 757 articles, we arrived at several key findings related to UK national newspapers' representations of childhood obesity. Coverage of the issue grew steadily from two articles in 1996 to a high of 82 articles in 2008, after which article frequency declined to 45 in 2009, before rising to a second peak of 69 in 2014. Childhood obesity was predominantly characterised as driven by individual-level factors, particularly parenting, dietary behaviours and inactivity, though societal drivers such as marketing were also identified. Similarly, there was greater focus on individual-level solutions than societal-level solutions. Societal constructions of the drivers of, and solutions to, obesity, were significantly more frequent within centre-left publications than centre right. Analysis

of time trends provided evidence of a small shift towards societal conceptualisations, with mentions of social solutions outnumbering individual solutions throughout the latter half of the sample period. Childhood obesity was frequently defined as a health risk in approximately half of articles, and was associated with females substantially more frequently than males, but more nuanced coding of gender representation in these articles is required.

Centre-left publications' greater focus on societal constructions of the causes of, and solutions to childhood obesity, and on the societal and health service burdens of childhood obesity, are in line with the communal and individual framings associated with left-wing and right-wing political ideologies. Entman describes the core process of building frames as '[selecting] some aspects of a perceived reality and [making] them more salient',[16] and this process is evident in UK newspaper representations of childhood obesity, with centre-left publications building frames that incorporate societal aspects of the childhood obesity problem, while centre-right publications omit them.

This research comprised a systematic analysis of a large sample of nineteen years of UK national newspaper coverage, facilitating statistical understandings of media frames of childhood obesity, including definitions, drivers and solutions. However, the research is subject to some limitations. The *Nexis* database does not archive articles from the *Daily Telegraph and Sunday Telegraph* prior to October 2000 and November 2000, respectively. However, the low frequency of reporting on childhood obesity prior to 2000 in the other sources in the sample during those years suggests that the absence of those two sources is unlikely to have had a relevant impact on our analysis. The method allowed quantitative analysis of media frames across a large sample, but not the nuanced analysis of specific aspects of framing that qualitative analysis would permit.

The coding frame was extensive, but subject to certain limitations. Coding did not record the types of issues discussed by each articles, which may have been valuable given the variety of different perspectives from which the issue may be viewed. Further, while mentions of males and females in relation to obesity in children were coded, coding did not differentiate between mentions of boys with obesity, girls with obesity, male parents and female parents. Given the frequently gendered nature of societal discourse about obesity, future research may benefit from analysing gendered representations of both children and parents within news coverage of childhood obesity. Additionally, future research may benefit from widening the search scope from childhood obesity to also cover childhood overweight. Our search terms were used to replicate those in a previous study[54] as therefore do not include the term 'childhood', which could lead to some relevant articles being missed. However, test searches suggest that incorporating the term 'childhood' into the search string returns negligible additional articles from UK national newspapers, so it is unlikely that those absent

articles would have substantially affected the analysis. Further limitations of the research stem from decisions made about the type of content analysed. The sole focus on article text was at the cost of analysing images, which have been found to be an important aspect of media representations of obesity.[23 31 55 56] Further, the focus on newspaper content was at the expense of data from other news sources, such as television and online news, or alternative sources, such as reader comments or social media posts. We argue that our focus on the evolution of the debate over time is not well suited to the rapidly changing online news environment, but acknowledge that incorporating other types of source could be valuable, as representations of childhood obesity have been found to vary by medium in the USA.[34] Finally, while links between media representations and public perceptions are well established, content analysis can only describe content, not determine how that content is received by audiences.

This research built on prior research examining media framing of general obesity[21] by extending the time period covered, taking a sole focus on childhood obesity and comparing coverage of childhood obesity to that of obesity in general. As would be expected, the growth in coverage of childhood obesity from 1996 to 2008 identified in our prior research[21] was replicated in the present research, but it was found that the rise did not continue beyond 2008, although it remained at an elevated level of coverage relative to pre-2002. Further research might investigate whether the increase in article frequency in the final year of the study period is indicative of a prolonged rise in coverage beyond 2014. Although it is likely that coverage of childhood obesity in 2007–2008 was elevated due to dissemination of, and activities related to, the UK government's Foresight report Reducing obesity: future choices, published in October 2007,[8] this trend mirrors that found in Barry *et al*[34] content analysis of US television and print news coverage of childhood obesity suggesting that, despite locally relevant policy events, trends in coverage of childhood obesity may follow transnational patterns. Barry *et al*[34] suggest that the decline in coverage may be an example of Downs'[57] 'issue-attention cycle', in which public attention to a specific issue will inevitably decline regardless of whether that issue reaches any conclusion. However, one area where our findings depart from those of Barry *et al*[34] is in individual and structural causes of childhood obesity, which they found to be equally frequent within the newspaper articles in their sample.

Both the original study by Hilton *et al* and the present study present some evidence of a shift away from a focus on individual constructions of drivers and solutions across their respective time periods. However, comparison of the two pieces of research suggests that, in comparison to general obesity, media frames of childhood obesity have a greater tendency to attribute responsibility to individuals. The disproportionate individual-level framing of childhood obesity might be explained by the presence of parents as mediators between children and public

policy. While children are vulnerable to societal and environmental pressures, and are often publicly viewed as deserving of legislative protection,[58–60] public discourse around childhood obesity may attribute greater individual responsibility to parents.[61] Hawkins and Linvill found that US news frequently identifies parents as both responsible for, and responsible for addressing, children's obesity, and conclude that this framing represents an obstacle to stimulating demand for a public policy response to the problem.[35] Boero's qualitative analysis of US media representations of childhood obesity identifies parents, and particularly mothers, as being 'under fire' for failing to foster healthy behaviours in their children.[28] Unlike in debates around unhealthy phenomena such as exposure to secondhand smoke, in which an adult lifestyle product may be perceived as unfairly invading children's spaces, feeding children occupies a complex position of being nurturing and essential, while also being a potential source of long-term health harms.[61]

For media content to drive public appetite for policy solutions to childhood obesity, media must both raise perceptions of the issue, through heightened coverage, and frame the issue as one demanding societal level, rather than solely individual level, solutions. Our research demonstrates that, while the salience of childhood obesity in UK national newspapers rose steadily from 1996 to 2008, that level of attention was not maintained subsequent to 2008, although there is reason to suggest that this may change in 2017/2018 with media coverage of the incoming levy on sugar-sweetened beverages in the UK.[62] While this faltering frequency of reporting may be undesirable for raising public consciousness, our analysis suggests that the frames constructed within those later years were characterised by a predominance of social solutions over individual solutions, which, if internalised by audiences, may stimulate public appetites for engaging the problem at the public policy level. Notably, this shift from individual to social framing occurred despite the well-documented complications caused by parents' roles as mediators between public policy and children's health behaviours. Taking these key findings into account, this study supports a mixed view of UK media framing of childhood obesity, in which positive changes in framing may be undermined by a decrease in salience. Those advocating for public policy responses to childhood obesity may seek to raise the issue's media profile, while continuing to promote social framings.

**Acknowledgements** The authors would like to thank Jonathan Olsen of the MRC/CSO Social and Public Health Sciences Unit, University of Glasgow, for his advice regarding the conduct of multiple logistic regression analysis. We would also like to acknowledge the contribution of Marie Löf and Mark Daku to the design of this study.

**Contributors** SH, CP and AN: study planning and conceptualisation; AN and CP: data coding; AN and CP: data analysis; AN and CP: drafting manuscript; SH: critical review of the manuscript.

**Funding** AN, CP and SH's time for this research was funded by the Informing Healthy Public Policy programme (MC_UU_12017-15 and SPHSU15) of the MRC/CSO Social and Public Health Sciences Unit, University of Glasgow.

**Disclaimer** The funding bodies had no role in the design, collection, analysis or interpretation of this study.

**Competing interests** None declared.

**Patient consent for publication** Not required.

**Provenance and peer review** Not commissioned; externally peer reviewed.

**Data sharing statement** Data were accessed from the *Nexis* newspaper database at https://www.nexis.com.

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
