## [Reviewer comments · BMJ Open]

ARTICLE DETAILS

TITLE (PROVISIONAL)	Media framing of childhood obesity: a content analysis of UK newspapers from 1996-2014
AUTHORS	Nimegeer, Amy; Patterson, Chris; Hilton, Shona

VERSION 1 - REVIEW

REVIEWER	Laura Nixon Berkeley Media Studies Group, United States
REVIEW RETURNED	08-Sep-2018

GENERAL COMMENTS	This study looks at nearly 20 years of UK news coverage about childhood obesity, and provides a number of insights into the framing of the issue over the time period analyzed. Overall, I think that the paper makes a strong addition to the literature about news framing, and the framing of childhood obesity in particular. However, I think that minor revisions are needed, mainly to clarify some methodological questions, and to adequately acknowledge the limitations of the study: Methods Pg. 9 – I have a number of questions about the headline tone variable. The literature review in the introduction provides a compelling basis for analyzing definitions of the problem, drivers, and solutions, but it's not clear to me why the authors were interested in the tone of the headline – i.e. how does it fit into their theoretical framework, has it been used in other framing studies, etc.? It could also be helpful to include examples of the three types of headlines. In addition, it wasn't clear to me whether coding agreement was calculated for headline tone - it does not seem to be included in Table 2. It would also be helpful to clarify/give an example of what constituted a mention of women and/or girls or men and/or boys. Results Pg. 13 – I was a little confused by the discussion about the specific drivers that centre-left publications were more likely to mention. The preceding sentence states that centre-left publications were more likely to mention societal drivers, but then two out of the three specific drivers more likely to be mentioned by centre-left publications are individual drivers. I think it would be clearer to first list the societal drivers more likely to be mentioned by centre-left publications, and then discuss differences in mentions of individual drivers by political alignment.
---

	Discussion Pg. 14 The second sentence in the Discussion section describes the changes in volume of coverage over time as follows: "Coverage of the issue grew steadily from 1996 to 2008, followed by a period of relatively infrequent coverage." To me, this description could imply that after 2008, coverage dropped back to 1996 levels, but that's not the case. I think it's important to acknowledge that the volume of coverage declined from its peak in 2008, but was still much higher than coverage levels pre-2002. Pg. 15/16 The authors state that observed trends in data could be due to inconsistencies in the Nexis database, and they mention the possibility of gaps in specific publications' archives. The Nexis database provides information about the dates of coverage for each publication's archive. The authors could check each publication in the Nexis database, and if there are in fact gaps in their archives in the time period studied, this should be included in the methods. In addition, I think it is important for the authors to acknowledge the potential limitations of their search strategy. The study searched for keywords in headlines in order to identify articles about childhood obesity. Given the volume of articles that mention keywords related to childhood obesity, this is a reasonable strategy to limit the volume of articles to analyze. However, I think it would be helpful to acknowledge in the Discussion section that this approach limits the analysis to articles that discuss the issue in depth, and doesn't capture instances where childhood obesity is discussed, but is not the main topic of the article. Finally, the search terms used are robust, but not exhaustive. In particular, the omission of a "childhood" term could lead to some relevant articles not being included. This should either be acknowledged as a limitation, or addressed in some way. Something that might be helpful would be to take a random sample of articles that would be pulled in with the "childhood" term, and assess whether they substantively differ from the existing universe of articles. Pg. 17 The authors first speculate that the reduction in coverage post-2008 could be due to the dissemination of a UK Government report, but then in a subsequent paragraph they note that an the trend in coverage echoes what has been found in coverage in the US. I think it would be clearer to have the discussion about the possible drivers of the changes in coverage over time together in one paragraph.
--	--

REVIEWER	Wen-ying Sylvia Chou National Cancer Institute, USA
REVIEW RETURNED	10-Sep-2018

GENERAL COMMENTS	This study presents findings from a content analysis of UK newspaper coverage of childhood obesity. A lot of work went into sampling and data coding and analysis, but ultimately, I am not clear about this study's scientific and public health impact. I have some specific comments:
--

	1) The abstract does not mention political alignments of the selected newspaper, a critical aspect of the analysis. Great alignment of abstract and actual study is needed. 2) The literature review focuses solely on journalism and its role in agenda-setting. I think it's important to also account for public discourse/perceptions such as obesity-related discussions on social media or on comments to a particular journalistic piece. There are many publications on obesity-related discussions on social media. The background section should incorporate literature on the changing media/journalism landscape. 3) The research goals are clearly described and Methods are generally sound. 4) How was the sampling decision made about the 11 newspapers? Were they ones with the highest circulation volume? How about the aim to strike a balance of political alignments? The authors could describe this decision more carefully on Page 8. Similarly, they did not specify what's considered not "relevant" (top of Page 9) as this can be very subjective and an example or two would be helpful to illustrate this decision process. 5) The reference that the sample was contemporaneous to the release of the UK Government Foresight project report lacks meaning since they did not ascertain how much of the analyzed articles covered this report or related matters. It's an answerable question given such a defined sample, so additional data extraction to ascertain a connection between news coverage and the report release is potentially feasible to report on this connection more fully. 6) The analysis of headline tone makes me wonder: should it depend on the very topic being covered? For example, the trend of increasing obesity rate can be indeed alarming and not reassuring, and hence an alarmist tone is justifiable. Also, this analysis does not tell us about the way each newspaper functions generally: Are some simply more inclined to use an alarmist tone than others? 7) typo on the second p value on Page 13 (not 0.046, but 0.000?) 8) I am not sure if the comparison between obesity and childhood obesity makes sense--could the later be included in the former but not vice versa? Did they have to be mutually exclusive? 9) Lastly, the authors hold the assumption that addressing childhood obesity requires (journalist) framing it as a public policy/societal issue instead of individual. While I generally agree with this public health perspective, it is an empirical question as to how much the individual- vs. collective/policy-level -causes and solutions can make an impact on obesity-related outcomes. Studying newspaper coverage alone does not tell us which (combination of) framing is most optimal in affecting the needed change to stop the growing rate of childhood obesity.
--	--

VERSION 1 – AUTHOR RESPONSE

REVIEWER ONE	
Pg. 9 – I have a number of questions about the headline tone variable. The literature review in the introduction provides a compelling basis for	

analyzing definitions of the problem, drivers, and solutions, but it's not clear to me why the authors were interested in the tone of the headline – i.e. how does it fit into their theoretical framework, has it been used in other framing studies, etc.? It could also be helpful to include examples of the three types of headlines. In addition, it wasn't clear to me whether coding agreement was calculated for headline tone - it does not seem to be included in Table 2.	On reflection, analysis of headline tone does not fit well within our analytical framework, and that the paper would be more elegant without it. As such, we have removed all content related to headline tone to produce, and are satisfied that manuscript is more focused as a result.
It would also be helpful to clarify/give an example of what constituted a mention of women and/or girls or men and/or boys.	This has now been added on page 9.
Results Pg. 13 – I was a little confused by the discussion about the specific drivers that centre-left publications were more likely to mention. The preceding sentence states that centre-left publications were more likely to mention societal drivers, but then two out of the three specific drivers more likely to be mentioned by centre-left publications are individual drivers. I think it would be clearer to first list the societal drivers more likely to be mentioned by centre-left publications, and then	We agree that this was confusingly worded in the initial draft and have now amended this section for clarity.

discuss differences in mentions of individual drivers by political alignment.	
Discussion Pg. 14 The second sentence in the Discussion section describes the changes in volume of coverage over time as follows: "Coverage of the issue grew steadily from 1996 to 2008, followed by a period of relatively infrequent coverage." To me, this description could imply that after 2008, coverage dropped back to 1996 levels, but that's not the case.	We have amended the text on pg 14 for clarity.
I think it's important to acknowledge that the volume of coverage declined from its peak in 2008, but was still much higher than coverage levels pre-2002.	Agree and we have added a statement to this effect in the Discussion on page 17.
Pg. 15/16 The authors state that observed trends in data could be due to inconsistencies in the Nexis database, and they mention the possibility of gaps in specific publications' archives. The Nexis database provides information about the dates of coverage for each publication's archive. The authors could check each publication in the Nexis database, and if there are in fact gaps in their archives in the time period studied, this	We thank the reviewer for reminding us about Nexis' archive gaps. All sources included in the research had complete archives in the archive, with the exceptions of the Telegraph and Sunday Telegraph, which were not archived prior to 2000. We feel satisfied that those sources' absence will not have had a substantial impact on the data given that there was so little coverage of the topic within the other sources during that early time period. However, it is important that we reflect on this, and as such we have added relevant content to the Methods and Discussion sections.

should be included in the methods.	
In addition, I think it is important for the authors to acknowledge the potential limitations of their search strategy. The study searched for keywords in headlines in order to identify articles about childhood obesity. Given the volume of articles that mention keywords related to childhood obesity, this is a reasonable strategy to limit the volume of articles to analyze. However, I think it would be helpful to acknowledge in the Discussion section that this approach limits the analysis to articles that discuss the issue in depth, and doesn't capture instances where childhood obesity is discussed, but is not the main topic of the article.	We have added an additional section about the limitations of this approach to the discussion section.
Finally, the search terms used are robust, but not exhaustive. In particular, the omission of a "childhood" term could lead to some relevant articles not being included. This should either be acknowledged as a limitation, or addressed in some way. Something that might be helpful would be to take a random sample of articles that would be pulled in with the "childhood" term, and assess whether they substantively differ from	We acknowledge that omitting the term 'childhood' is an oversight that could potentially affect the sample. Fortunately, having run some test searches, adding 'childhood' to the search string does not appear to make a substantial difference. For example, searching the previous year of Nexis' 'UK National Newspapers' source list returns 54 articles without 'childhood', and 56 articles with it. The search terms were initially generated by translating those used in an analysis of Swedish newspaper representations of childhood obesity, and the absence of 'childhood' from the resulting search terms is a regrettable, albeit not significant, oversight. We have added a sentence to the limitations paragraph section to address the absence of this search term.

the existing universe of articles.	
Pg. 17 The authors first speculate that the reduction in coverage post-2008 could be due to the dissemination of a UK Government report, but then in a subsequent paragraph they note that an the trend in coverage echoes what has been found in coverage in the US. I think it would be clearer to have the discussion about the possible drivers of the changes in coverage over time together in one paragraph.	This section has been amended.
REVIEWER TWO	
1) The abstract does not mention political alignments of the selected newspaper, a critical aspect of the analysis. Great alignment of abstract and actual study is needed.	We have amended the abstract to be clearer that we analysed the data by political alignment.
2) The literature review focuses solely on journalism and its role in agenda-setting. I think it's important to also account for public discourse/perceptions such as obesity-related discussions on social media or on comments to a particular journalistic piece. There are many publications on obesity-related discussions on social media. The background section should incorporate literature on the changing	We agree that discussion of new media was lacking, therefore we have amended the introduction to acknowledge social media research on obesity and further describe the changing media landscape. We do also acknowledge the limitation of the research as focused on traditional print news media on page 16. We have also added a new paragraph to the introduction that acknowledges non-news and social media research and the changing media landscape, but also further justifies our use of traditional news media.

media/journalism landscape.	
3) The research goals are clearly described and Methods are generally sound.	We thank the reviewer for their kind feedback.
4) How was the sampling decision made about the 11 newspapers? Were they ones with the highest circulation volume? How about the aim to strike a balance of political alignments? The authors could describe this decision more carefully on Page 8. Similarly, they did not specify what's considered not "relevant" (top of Page 9) as this can be very subjective and an example or two would be helpful to illustrate this decision process.	We have added to this section, clarifying our sampling process, as well as our exclusion criteria. We also added a line on the inherent subjectivity of this type of content analysis work.
5) The reference that the sample was contemporaneous to the release of the UK Government Foresight project report lacks meaning since they did not ascertain how much of the analyzed articles covered this report or related matters. It's an answerable question given such a defined sample, so additional data extraction to ascertain a connection between news coverage and the report release is potentially feasible to report on this connection more fully.	We agree with this to a certain extent – it would be possible to go back and re-code the data to find the proportion of articles that mention the foresight report, however, we would argue that what we cannot account for is how many articles were stimulated due to the increased attention brought to the topic by the foresight report, but which may not have directly referenced it. In other words, the publication of the report and subsequent coverage brought the issue to public attention, which stimulated more reporting on the topic, and we can only surmise about journalistic intent in this matter.
6) The analysis of headline tone makes me wonder: should it depend	

on the very topic being covered? For example, the trend of increasing obesity rate can be indeed alarming and not reassuring, and hence an alarmist tone is justifiable. Also, this analysis does not tell us about the way each newspaper functions generally: Are some simply more inclined to use an alarmist tone than others?	On reflection, analysis of headline tone does not fit well within our analytical framework, and that the paper would be more elegant without it. As such, we have removed all content related to headline tone to produce, and are satisfied that manuscript is more focused as a result.
7) typo on the second p value on Page 13 (not 0.046, but 0.000?)	This has been corrected.
8) I am not sure if the comparison between obesity and childhood obesity makes sense-- could the later be included in the former but not vice versa? Did they have to be mutually exclusive?	The 2012 study data that we have used for this comparison included any coverage of obesity in general (including adult and child), but each article within that sample was coded to indicate whether it contained discussion of obesity in children or not. As a result, we were able to remove those articles from the 2012 sample that discussed children, leaving only articles about adult (or 'general') obesity for comparison with our new sample., and allowing comparison of two mutually exclusive samples (i.e. reporting on adult obesity 1996-2010, and reporting on child obesity 1996-2014).
9) Lastly, the authors hold the assumption that addressing childhood obesity requires (journalist) framing it as a public policy/societal issue instead of individual. While I generally agree with this public health perspective, it is an empirical question as to how much the individual- vs. collective/policy-level - causes and solutions can make an impact on obesity-related outcomes. Studying newspaper coverage alone does not tell us which (combination of)	We agree that solving the problem demands a certain balance of individual and collective approaches, and that our analysis cannot shed light on what that balance is. However, our research is guided by two contextual realities: firstly that research has demonstrated that media representations of obesity has promoted a predominantly individual-level framing, and secondly that obesity researchers and public health professionals have long worked to emphasise the need for policy-level approaches within a multi-level package of solutions. For example, on page 5 we write that: “Childhood obesity is a complex problem, with a complex set of drivers and potential solutions ranging from the individual to the environmental(8). Ebbeling and colleagues (1) identify a wide range of causes, but argue that the problem “can be primarily attributed to adverse environmental factors”, and identify a need for “straightforward, if politically difficult” solutions spanning homes, schools, the built environment, health care, marketing, media and politics. This multi-level package of solutions echoes Friedman’s

framing is most optimal in affecting the needed change to stop the growing rate of childhood obesity.	assertion that a 'full-court press' targeting 'every dimension of the problem' is necessary(9)." We acknowledge that for the media to focus solely on societal solutions without ignoring individual factors would be undesirable, but our position that a relatively greater focus on societal solutions is desirable is based on evidence-based understandings that media currently frame obesity as a predominantly individual issue, and that the current best thinking on tackling the obesity problem instead emphasises societal and environmental change.
--	---

VERSION 2 – REVIEW

REVIEWER	Laura Nixon Berkeley Media Studies Group, United States
REVIEW RETURNED	11-Jan-2019

GENERAL COMMENTS	The authors have done a great job of addressing my concerns, clarifying methods where necessary, etc.
--

REVIEWER	Wen-ying Sylvia Chou National Cancer Institute, USA
REVIEW RETURNED	12-Feb-2019

GENERAL COMMENTS	The revision is quite responsive to my earlier comments. Just a few very minor comments:  1) Ref 42 (So, Prestin, et al.) is based on a larger study/publication which is a more appropriate and comprehensive as a citation to support the authors' argument: Chou, Prestin, et al. Obesity in Social Media (https://www.ncbi.nlm.nih.gov/pubmed/25264470) 2) Removing tone from the analysis makes the study clearer. 3) pages 7-8: there's growing evidence the news media have to react to discussions on social media, so in this case the interplay between "traditional" and "new" media platforms is evolving and highly connected. The author makes an arbitrary contrast between the two, when I believe they are highly interconnected and the lines quite blurry (consider 24-hour news cycle and comments posted to respond to a traditional news piece, and the viral spread of a piece of information necessitates a journalist to cover a story). I would revise this paragraph to reflect this intertwined nature and still validate the importance of this analysis.
---

VERSION 2 – AUTHOR RESPONSE

We thank the reviewers for the time and care they have put into reviewing our paper, we believe it has made the manuscript much stronger.

Reviewer 1: We thank the reviewer for their kind comment.

Reviewer 2:

1) We have changed this reference.

2) We thank the reviewer for their kind comment.

3) We completely agree and have revised the paragraph on pages 7-8 to expand on the complex relationship between traditional and new media platforms.